# Submesoscale transition from geostrophic flows to internal waves in the northwestern Pacific upper ocean

Bo Qiu[1], Toshiya Nakano[2], Shuiming Chen[1] & Patrice Klein[3,†]

With radar interferometry, the next-generation Surface Water and Ocean Topography satellite mission will improve the measured sea surface height resolution down to 15 km, allowing us to investigate for the first time the global upper ocean variability at the submesoscale range. Here, by analysing shipboard Acoustic Doppler Current Profiler measurements along 137°E in the northwest Pacific of 2004–2016, we show that the observed upper ocean velocities are comprised of balanced geostrophic flows and unbalanced internal waves. The transition length scale, $L_t$, separating these two motions, is found to depend strongly on the energy level of local mesoscale eddy variability. In the eddy-abundant western boundary current region of Kuroshio, $L_t$ can be shorter than 15 km, whereas $L_t$ exceeds 200 km along the path of relatively stable North Equatorial Current. Judicious separation between the geostrophic and internal wave signals represents both a challenge and an opportunity for the Surface Water and Ocean Topography mission.

[1] Department of Oceanography, University of Hawaii at Manoa, 1000 Pope Road, Honolulu, Hawaii 96822, USA. [2] Global Environment and Marine Department, Japan Meteorological Agency, 1-3-4 Otemachi, Chiyoda-ku, Tokyo 100-8122, Japan. [3] Laboratoire d'Océanographie Physique et Spatiale, Ifremer/CNRS/UBO/IRD, Plouzane 29280, France. † Present address: Environmental Science and Engineering, Caltech and J.P.L. (NASA), Pasadena, USA. Correspondence and requests for materials should be addressed to B.Q. (email: bo@soest.hawaii.edu).

Accumulation of the satellite altimeter-derived sea surface height (SSH) data of the past 25 years has significantly improved our understanding of the mesoscale variability of the upper ocean[1–3]. A critical limitation of the nadir-looking altimeters is its 100–300-km spacing between the satellite ground tracks. Even with data merged from multiple altimeters, the spatial resolution in a two-dimensional SSH map is typically on the order of 150 km in wavelength[3,4]. This resolution is inadequate to fully capture the mesoscale oceanic signals that contain 90% of the kinetic energy of the ocean[5] and misses completely the submesoscale oceanic features that have length scales of 10–150 km. Upper-ocean submesoscale processes are dynamically important because they determine the equilibrium state of the upper ocean through the turbulent kinetic energy cascade and energy dissipation[5]. They are also crucial to how the surface ocean communicates with the subsurface interior ocean, affecting the mixed layer development and upper-ocean thermal anomalies[6–8]. In addition to the physical properties, upper-ocean, meso- and submesoscale processes impact the $CO_2$ uptake, nutrient supply and biogeochemistry of the upper ocean as well[9–11].

Shipboard Acoustic Doppler Current Profiler (ADCP) measurements are an effective means to observe the mesoscale and submesoscale upper ocean currents[12]. With research vessels that sail at a typical speed of $\sim 5\,\mathrm{m\,s^{-1}}$, the absolute velocity signals resolved by the ADCPs have a spatial resolution of a few kilometres. An early study based on the ADCP data across the Gulf Stream provided statistical descriptions on the along-track wavenumber spectra of velocity variance and their comparisons with the surface geostrophic flows from the along-track SSH measurements from altimeters[13]. More recent studies utilizing the ADCP data have taken advantage of contemporaneous along-track and cross-track velocity information to determine through spectral Helmholtz decomposition the rotational versus divergent components of the upper ocean velocity field[14–16]. By effectively separating the balanced geostrophic flows from those of the unbalanced wave motions that tend to flatten the wavenumber spectra, these recent studies are able to obtain more reliable depictions about the turbulent nature of the upper ocean circulation at several locations of the world ocean. A consensus resulting from these studies is that the upper ocean geostrophic variability in the mesoscale-submesoscale range shorter than the instability-forced length scales is dominated more by the interior quasi-geostrophic (QG) dynamics[17] than by the surface QG dynamics[18], and that the flattening in wavenumber spectral slopes is because of the occurrence of mixed layer instability.

In the northwest Pacific, repeat shipboard ADCP surveys have been conducted by Japan Meteorological Agency (JMA) along the 137°E meridian on the semi-annual or seasonal basis in the past two decades[19,20] (see the Methods section for more details of the JMA ADCP data). As shown in Fig. 1a, the 137°E section from 3°N to 34°N traverses four distinct dynamical regimes in which the background mean circulation and the mesoscale eddy variability exhibit different characteristics. North of 28°N exists the deep-reaching western boundary current, the Kuroshio, and its southern and northern recirculating flows[21,22] (Fig. 2a). Mesoscale eddy variability in this northern latitude band is dominated by the intrinsic Kuroshio variability that can reach deep below the main thermocline (Fig. 2b). The 17–26°N band in the southern half of the wind-driven subtropical gyre sees the presence of multiple surface trapped eastward jets, known as the Subtropical Countercurrents (STCCs)[23,24]. Within this band, enhanced mesoscale eddy variability is generated by baroclinic instability of the vertically sheared, eastward-flowing STCC and westward-flowing North Equatorial Current

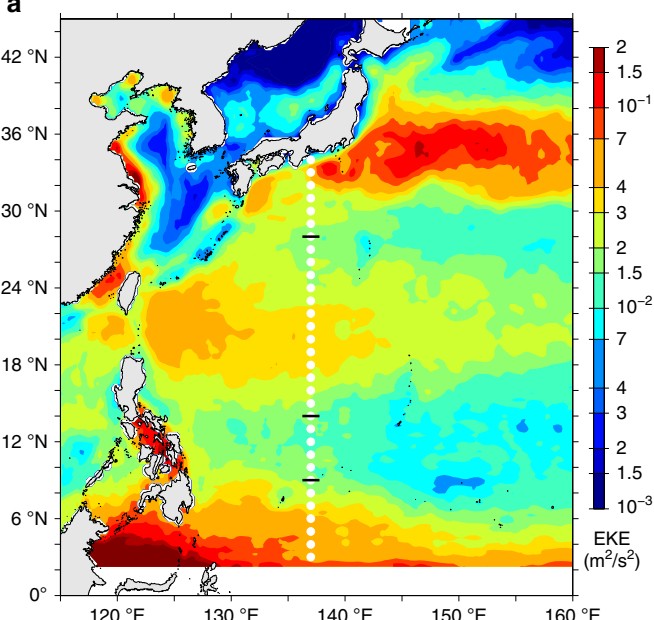

**a**

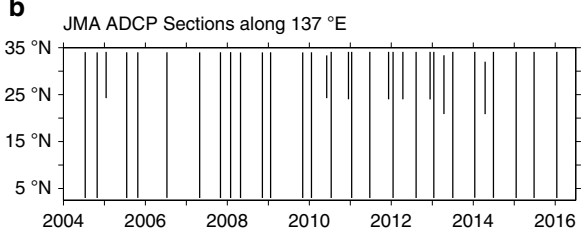

**b**

JMA ADCP Sections along 137 °E

**Figure 1 | Regional eddy variability and ADCP track.** (**a**) Surface eddy kinetic energy distribution in the northwestern Pacific based on the weekly AVISO SSH anomaly data of 2004–2015. White dots along 137°E denote the track of repeat ship-board ADCP measurements by Japan Meteorological Agency. (**b**) Time-latitude plot of the 33 JMA cruises along 137°E used in the present study. Short black lines along 28°N, 14°N and 9°N demarcate the boundaries of the Kuroshio, STCC, NEC and NECC band.

(NEC) in the subsurface layer[25,26]. The time-mean westward-flowing NEC exists in the broad latitude band of 7–21°N (Fig. 2a). Within its main body of 9–14°N, the NEC is highly stable and this can be confirmed by both the available satellite altimeter data measurements (Fig. 1a) and the repeat ADCP surveys (Fig. 2b). Although the NEC is an intense zonal current, its dynamical stability is because of the lack of change in meridional gradient of potential vorticity in its time-mean flow structures[25,27]. The wind-driven North Equatorial Countercurrent (NECC) exists in the southern 3–7°N band. Like the Kuroshio in the northern end of the 137°E section, the NECC is an eastward-flowing return flow; it is dynamical unstable because of the inability of a wind-driven gyre to smoothly connect potential vorticity of the western boundary current (the Mindanao Current in the NECC case) back to the interior Sverdrup flow[28]. The high-mesoscale eddy variability detected in the 3–7°N band (Figs 1a and 2b) is shown recently to be caused by barotropic instability of the laterally sheared NECC (ref. 29).

Here we show that with the coverage of these dynamically differing circulation regimes, the multi-year ADCP measurements along 137°E provide a unique opportunity to explore the regionally and seasonally varying mesoscale and submesoscale upper ocean variability. By adopting the analytical tools

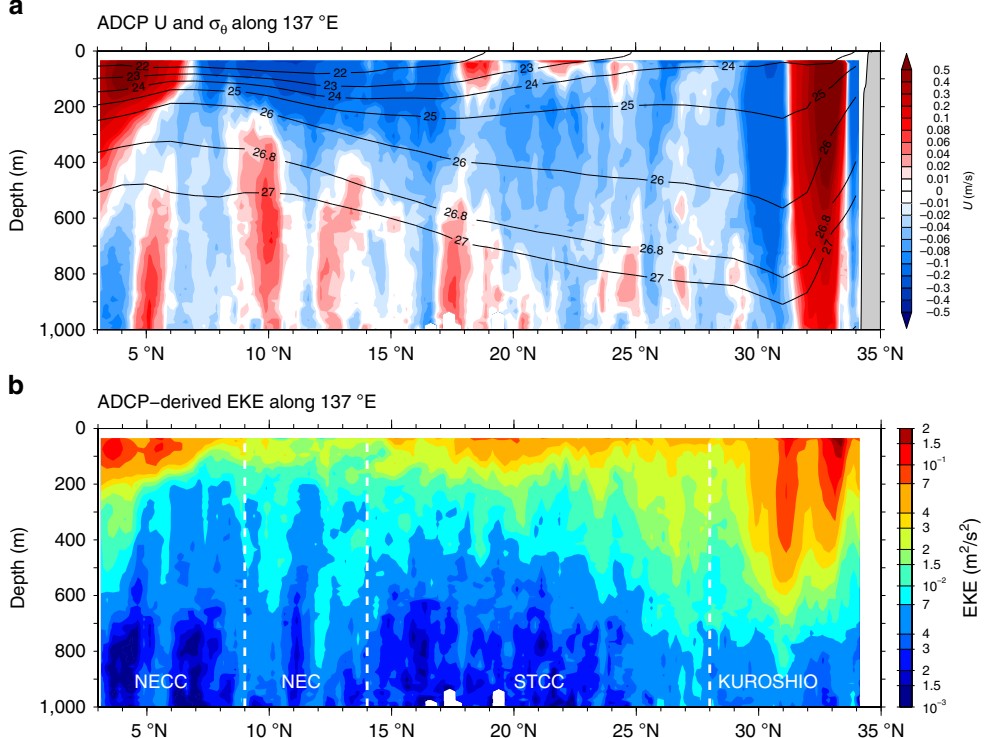

**Figure 2 | Mean and time-varying circulation along 137E.** Latitude-depth sections of (**a**) ADCP-derived zonal mean velocity (coloured contours) and CTD-derived density (black contours in $\sigma_\theta$) and (**b**) ADCP-derived eddy kinetic energy along 137°E from the JMA's repeat surveys. Here, eddy velocities are evaluated by removing the time-mean ADCP-derived velocities from individual cruises. Notice that contour scales in (**a**) are nonlinear and red (blue) colours denote eastward (westward) flows. White dashed lines delineate the four dynamical bands used in the spectral analyses.

of Helmholtz and wave-vortex decompositions[14–16], we find that a transition exists between the geostrophic- versus internal wave-dominant motions and that its length scale depends sensitively on the regional mesoscale energy level.

## Results

**Velocity data and decompositions.** Given the distinct dynamics and mesoscale eddy kinetic energy level in the different geographical regions, we divide the 137°E transect into 4 subregions in our analyses: the Kuroshio band of 28–34°N, the STCC band of 14–28°N, the NEC band of 9–14°N, and the NECC band of 3–9°N (see white dashed lines in Fig. 2b). Note that this division helps to ensure the spatial homogeneity of the regional eddy variability. Within each band, available cross-track (zonal) and along-track (meridional) velocity data are grouped into 500 km-long segments with 50% overlap, a Hanning window is applied to each segment, and a discrete Fourier transform is computed. The finalized along-track wavenumber spectrum for kinetic energy (KE) is evaluated based on the squares of Fourier components averaged over all available segments. In addition to the KE wavenumber spectra for zonal and meridional velocities, $\hat{C}^u/2$ and $\hat{C}^v/2$, we utilize the Helmholtz decomposition[15,16] and calculate the KE spectra for the rotational and divergent flow components of the velocity field: $\hat{K}^\psi$ and $\hat{K}^\phi$. By assuming the Garrett–Munk spectrum[30] for internal wave motions, we further evaluate the wavenumber spectra for the unbalanced wave motions versus the balanced vortex motions: $\hat{C}_W^u/2$ and $\hat{C}_W^v/2$ versus $\hat{C}_V^u/2$ and $\hat{C}_V^v/2$. Details of the velocity spectral decompositions are provided in the Methods section.

**The Kuroshio band.** Figure 3a shows that the ADCP-derived zonal velocity spectrum $\hat{C}^u/2$ has a $k^{-2.4}$ slope in the 10–200 km range, and is steeper than $\hat{C}^v/2$ across all resolved wavelengths. Assuming geostrophy, this observed spectral slope is consistent with the along-track SSH slopes ($\sim k^{-4}$) inferred from satellite altimetry measurments[31]. In this northern band, divergent flows have a much lower KE spectral level than that of the rotational flows (cf. green and red lines in Fig. 3a for $\hat{K}^\phi$ and $\hat{K}^\psi$, respectively). When decomposed into the balanced geostrophic motions (Fig. 3b), $\hat{C}_V^u/2$ follows closely a $-3$ power law ($k^{-2.6}$ by least-squared fitting in the 10–200 km range) and $\hat{C}_V^u/\hat{C}_V^v$ has a ratio close to 3 (2.8 ± 0.7). Both of these results suggest the eddy variability here is related to combined interior QG and mixed layer instabilities, and this combined nature of turbulence in the Kuroshio has been confirmed by a recent high-resolution circulation simulation of the North Pacific[32]. Compared with the balanced geostrophic motions, the internal waves make a less contribution to eddy signals longer than 20 km (cf. green line in Fig. 3c). The geostrophic flows and internal waves become comparable in spectral amplitude at the wavelength $L_t = \sim 15$ km (see the crossing between the green and red lines in Fig. 3c). Notice that a similar transition from geostrophic motions to inertia-gravity waves was observed in the midlatitude atmospheric wind KE spectrum[33].

**The STCC band.** Compared with the Kuroshio band, Fig. 3d shows that the observed zonal and meridional velocity spectra $\hat{C}^u/2$ and $\hat{C}^v/2$ in the STCC band have slopes closer to $-2$ (at $-2.3$ and $-2.0$, respectively) in the 10–200 km range. The balanced motion spectra presented in Fig. 3e, $\hat{C}_V^u/2 \sim k^{-2.6}$, $\hat{C}_V^v/2 \sim k^{-2.5}$ and $\hat{C}_V^u/\hat{C}_V^v = 2.3 \pm 0.3$, again fall

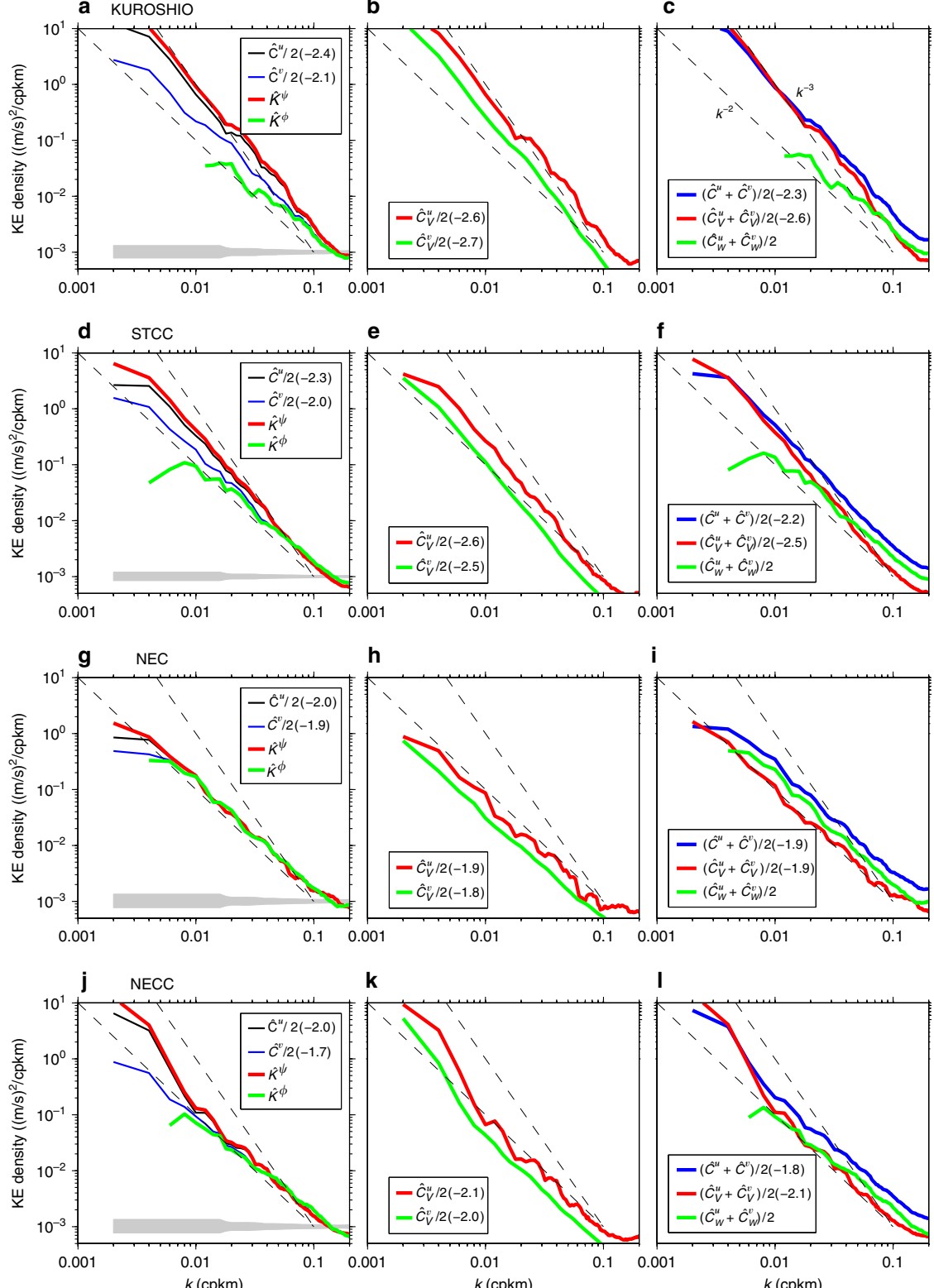

**Figure 3 | Spectra of velocity components in four dynamical bands.** Along-track wavenumber spectra for different velocity components in the (**a**–**c**) Kuroshio, (**d**–**f**) STCC, (**g**–**i**) NEC and (**j**–**l**) NECC band. (**a**,**d**,**g**,**j**): zonal, meridional, rotational and divergent velocity spectra. (**b**,**e**,**h**,**k**): zonal versus meridional geostrophic velocity spectra. (**c**,**f**,**i**,**l**): kinetic energy spectrum (blue) and its decomposition into wave (green) and geostrophic (red) motions. Values in parentheses indicate the best-fit spectral slopes in the 10–200 km range. Dashed lines in background denote the $k^{-2}$ and $k^{-3}$ reference curves and grey bands denote the 95% confidence intervals.

in between $k^{-3}$ and $k^{-2}$, the spectral slopes predicted for the interior QG turbulence and mixed layer instability, respectively[17,34]. Numerical simulation evidence supporting the co-existence of interior QG and mixed layer instabilities in the STCC can be found in ref. 35. Compared with the Kuroshio band, the internal wave motions in the STCC band expand to the longer wavelengths (Fig. 3f) and the geostrophic and internal wave motions become comparable in spectral amplitude at the wavelength $L_t = \sim 50$ km.

**The NEC band**. In this low mesoscale eddy band (recall Fig. 2b), Fig. 3g shows that there exists little difference between the cross-track and along-track velocity variance spectra. The same also appears to be true for the decomposed rotational and divergent flow spectra. Slopes for all spectra $\hat{C}^u/2$, $\hat{C}^v/2$, $\hat{K}^\phi$ and $\hat{K}^\psi$ in Fig. 3g have a flat value close to $-2$. For the balanced geostrophic motions, the spectral slopes again follow a $k^{-2}$ power law, but their energy levels are extremely low (Fig. 3h). In comparison, the internal wave motions in this band (the green line in Fig. 3i) have a spectral amplitude comparable to those in the Kuroshio and STCC bands. At all scales shorter than $L_t = \sim 250$ km, Fig. 3i reveals that the balanced motions have a spectral amplitude lower than that of the internal waves in the NEC band.

**The NECC band**. Like in the NEC band, $\hat{C}^u/2$, $\hat{C}^v/2$, $\hat{K}^\phi$ and $\hat{K}^\psi$ in the NECC band have flat spectral slopes close to $k^{-2}$ in the wavelength range shorter than $\sim 80$ km (Fig. 3j). The spectral slopes steepen for $\hat{C}^u/2$ and $\hat{K}^\psi$ at the longer wavelengths. Notice that the transition from geostrophic flows to internal waves also occurs at about 80 km (Fig. 3l).

**Seasonality**. To gain more dynamical insights into the geostrophic and internal wave motions in different geographical bands, we further investigate the KE spectra for these two motions from the 10 winter (Jan–Feb) and 10 summer (June–Aug) cruises. In the northern Kuroshio band (Fig. 4a–c), while there exists no clear difference in eddy kinetic energy level at wavelengths > 120 km, smaller-scale mesoscale/sub-mesoscale variability has a significantly larger energy level in winter than in summer. Although the internal wave motions have a larger energy level in winter than in summer, as shown in Fig. 4c, much of this seasonal difference is due to the geostrophic motions because they have much larger spectral amplitudes than those of the internal waves (cf. Fig. 4b,c). Because of the enhanced eddy variability at scales shorter than 100 km relating to the mixed layer instability[32], the spectral slope for the geostrophic motions is $-2.4$ in winter, as compared with $-3.0$ in summer. This suggests that the balanced eddy variability in the Kuroshio band in winter is controlled by combined interior QG and mixed layer instabilities, whereas that in summer is dictated by the interior QG dynamics alone. Similar seasonally varying behaviours of submesoscale eddy variability have been observed recently in the Gulf Stream region[36].

Along the STCC band (Fig. 4d,e), the seasonal spectral differences are quite similar to those detected in the neighbouring Kuroshio band with a higher energy level for both the total and balanced geostrophic motions in winter than in summer. The seasonal amplitude difference is smaller in the STCC band than the Kuroshio because the regional wintertime mixed layer instability is comparatively weaker in the STCC band[35]. Seasonal difference in internal wave motions in the STCC band is small (Fig. 4f) and because of the reduced KE level for the geostrophic motions, the transition scale $L_t$ is at $\sim 25$ km in winter versus $\sim 100$ km in summer.

In the NEC band (Fig. 4g–i), no appreciable seasonal differences are found in the total KE spectra between winter and summer. When decomposed, the geostrophic motions appear to have similar overall spectral amplitudes as the internal waves in winter. In summer, on the other hand, the geostrophic motions are observed to be overwhelmed by the internal waves. The enhanced summertime internal wave motions in the NEC band is likely caused by seasonal genesis of typhoons and tropical cyclones in the region[37]. In the NECC band equatorward of 9°N, Fig. 4j–l reveals that little seasonal differences exist between the winter and summer KE spectra of the original and decomposed velocities.

**Discussion**

From the decomposed KE spectra shown in (c,f,i,l) of Fig. 3, it is noticeable that while the energy level for the geostrophic motions can vary significantly depending on the background time-mean circulation, the energy level for internal waves remains fairly uniform across the four flow bands. As a result, the transition wavelength $L_t$ appears to be controlled more by the regional geostrophic than internal wave motions. To explore $L_t$'s dependence more explicitly, we plot in Fig. 5a,b the KE spectra as a function of latitude for the decomposed geostrophic and internal wave motions within each 500 km segment along 137°E. Reflecting the instability nature of the background time-mean circulation, the KE spectra for the geostrophic motions shown in Fig. 5a are clearly latitude-dependent. On the other hand, Fig. 5b reveals a relatively weak latitude-dependence in the KE spectra for internal waves. This weak dependence of the internal wave KE spectrum is consistent with the original saturation hypothesis of Garrett–Munk[30], and with a recent numerical study that suggests that wave-wave interactions generate a universal internal wave spectrum regardless of latitude and magnitude of near-inertial and internal tidal forcings[38].

The red lines in Fig. 5a,b indicate the transition scale $L_t$, at which the KE spectral level of the geostrophic motions matches that of the internal waves. The smallest $L_t$ value, < 10 km, is detected north of 28°N where the observed kinetic energy is the highest along 137°E (Fig. 5c) because of the intrinsic variability of the Kuroshio and its southern recirculation gyre[39]. The largest $L_t$ of > 200 km, on the other hand, is detected along the NEC path in the 10–13°N band where the observed KE has the lowest level. Lack of sign change in the meridional potential vorticity gradient within the NEC system has been argued in the past to suppress the instability[25,27]. Across the broad STCC band of 14–28°N and along the NECC path where the observed eddy variability level is moderate, Fig. 5c reveals that $L_t$ falls generally in between the 20 and 70 km range.

While showing a general negative correspondence between the local KE level and $L_t$ in Fig. 5c, the linear correlation coefficient between these two quantities is not very high: $r = -0.44$. Compared with the local KE level, we find that a better correspondence exists between $L_t$ and $L_e$, the local energy-containing length scale, defined by $L_e = \left[ \int kE(k)\mathrm{d}k / \int E(k)\mathrm{d}k \right]^{-1}$, where $E(k)$ is the KE spectrum and $k$ is the along-track wavenumber. As shown in Fig. 5d, $L_t$ and $L_e$ have a correlation coefficient $r = -0.56$. It will be interesting in future studies to verify if the same higher correlation between $L_t$ and $L_e$ (than between $L_t$ and local KE level) is detected in other parts of the world ocean and, if it is, the dynamical reason(s) behind this higher correlation between $L_t$ and $L_e$.

Because the ADCP measurements extend below the surface 100 m layer, we present in Fig. 6 the depth-dependence of the geostrophic and internal wave KE spectra, and the resultant

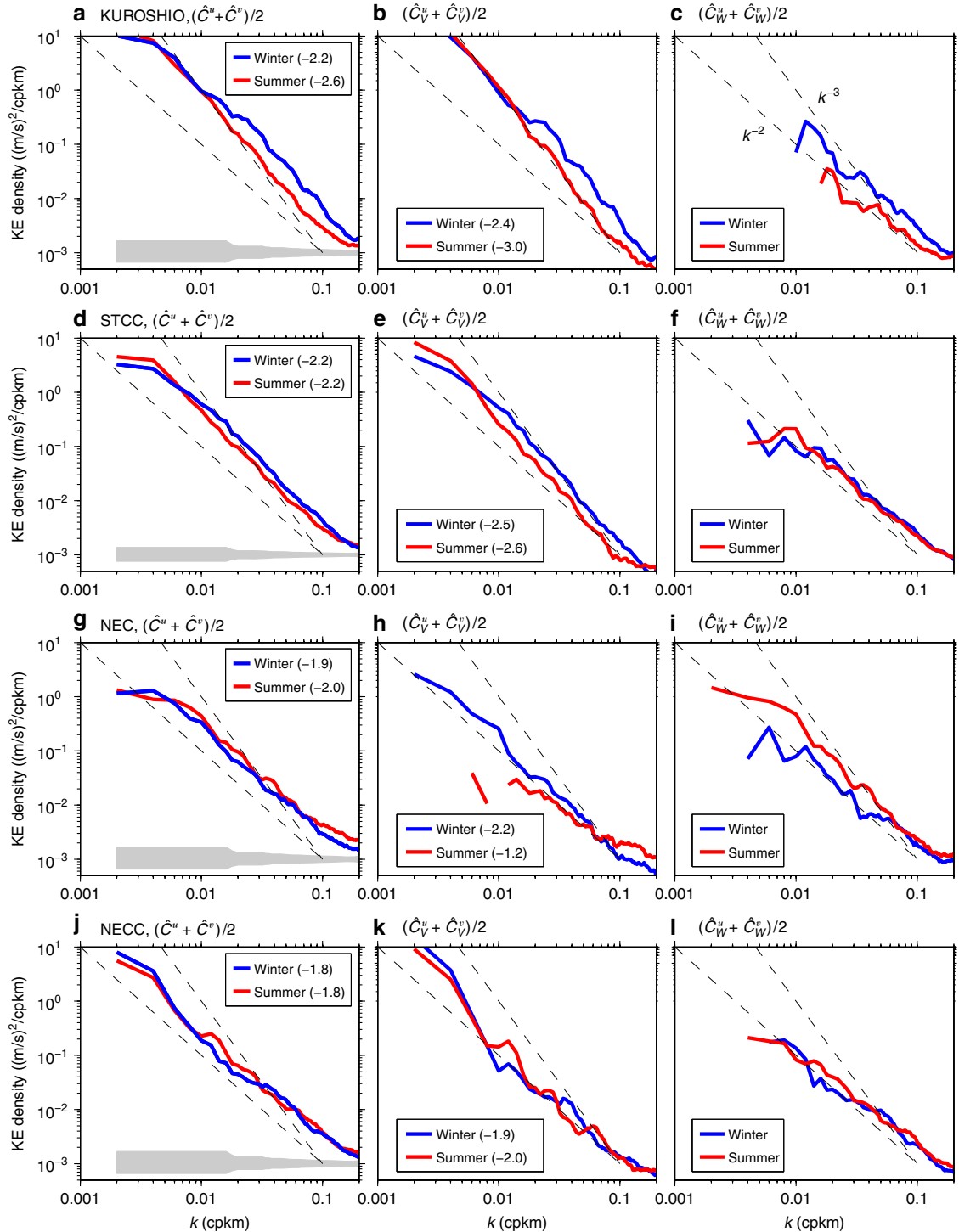

**Figure 4 | Spectra of velocity components in winter versus summer.** Winter versus summer kinetic energy spectra in the (**a–c**) Kuroshio, (**d–f**) STCC, (**g–i**) NEC and (**j–l**) NECC band. (**a,d,g,j**): total KE, (**b,e,h,k**): balanced motion KE and (**c,f,i,l**): internal wave KE. Dashed lines in background denote the $k^{-2}$ and $k^{-3}$ reference curves and grey bands denote the 95% confidence intervals.

transition scale, in the four flow regimes. In the northern band where the Kuroshio is deep-reaching (Fig. 2), Fig. 6a,b reveal that $L_t$ and the geostrophic and internal wave KE spectra all have a relatively weak dependence on depth. That the KE spectral slopes for geostrophic motions have a weak depth-dependence is inconsistent with the surface QG turbulence, whose theory predicts a systematic and rapid steepening of the KE spectrum with an increasing depth[14,40]. These slopes,

however, better agree with the argument that mesoscale eddies are captured by the first baroclinic mode[41]. In the STCC band where the mean flow is shallow, the geostrophic KE spectral level drops rapidly with depth (Fig. 6c). With the weak depth-dependence in the internal wave KE spectra, $L_t$ in the STCC band lengthens as the depth increases (Fig. 6d). In contrast to the Kuroshio and STCC bands, Fig. 6e,f indicate that the transition scale decreases with depth in the relatively stable

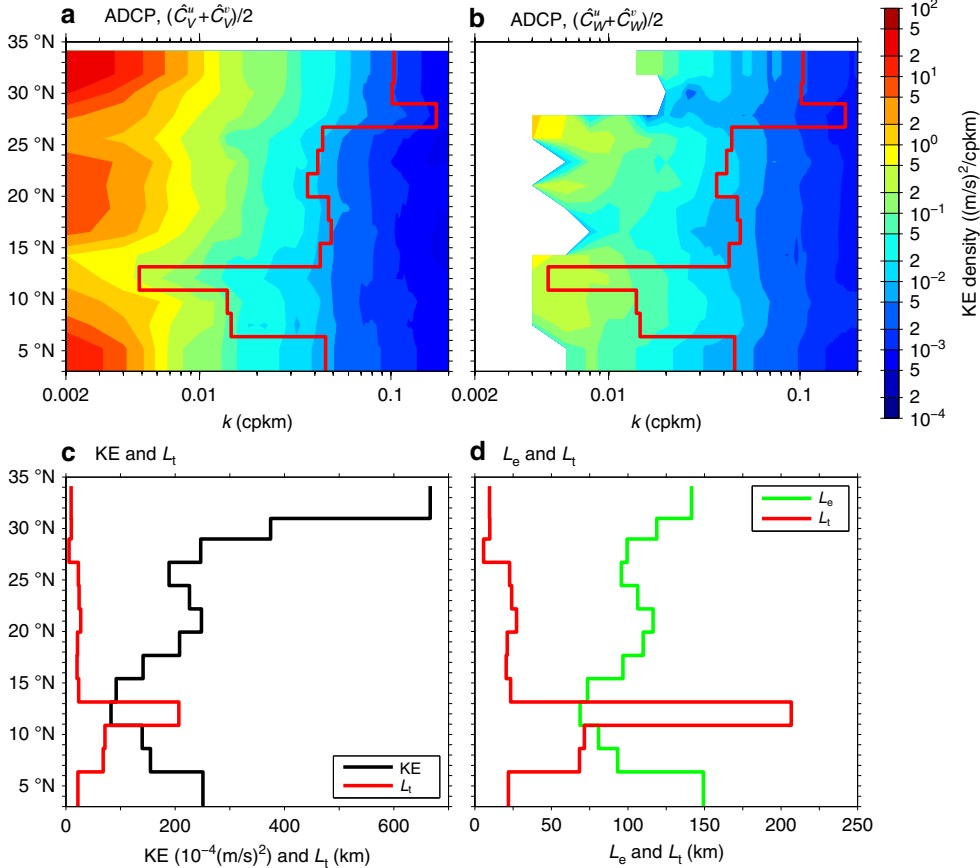

**Figure 5 | Transition scale dependence on latitude and eddy characteristics.** ADCP-derived kinetic energy (KE) spectra (in colour) for (**a**) balanced geostrophic motions versus (**b**) unbalanced wave motions estimated over 500 km-segments along 137°E as a function of latitude. Red lines denote the transition scale $L_t$ where the geostrophic motion spectral amplitude matches that of the internal waves. (**c**,**d**) show comparisons between $L_t$ and the total KE value and between $L_t$ and the energy-containing length scale $L_e$, respectively.

NEC band. This decrease in $L_t$ is in large part due to the enhanced geostrophic eddy variability that exists below the westward-flowing NEC and is associated with the sub-thermocline, eddy-forced, North Equatorial Undercurrent (NEUC) jets[42,43] (that is, the narrow eastward jets along 9°N, 13°N and 18°N shown in Fig. 2a). In the southernmost NECC band, $L_t$ appears to first increase below 100 m because of the decrease in the geostrophic motion KE level (Fig. 6g). Further below, at depth $>300$ m, the transition scale reverses to a decreasing trend with depth and this decrease in $L_t$ appears to be largely due to the weakening in internal wave KE level with depth (Fig. 6h).

That the $L_t$ value in the surface layer can vary geographically by $>10$-folds is highly relevant for the forthcoming SWOT mission, because $L_t$ likely delineates the threshold wavelength shorter than which the SWOT-measured SSH data may no longer be used to accurately infer the surface geostrophic velocities. The results of our analyses suggest that with its expected spectral resolution at 15 km (ref. 44), the internal wave motions should not pose a serious problem for the SWOT mission to detect the balanced submesoscale motions in mesoscale-rich regions, such as the western boundary currents and their extensions. In the regions of moderate mesoscale activities (for example, subtropical countercurrent bands and tropical western boundary current outflow regions), we recommend caution in diagnosing surface geostrophic velocity from SSH at scales smaller than about 70 km.

The real challenge for the SWOT mission comes when one attempts to retrieve the time-varying geostrophic flow signals in the low mesoscale variability regions that include not only the westward-flowing Sverdrup zonal flow bands in the tropics, but also the vast eastern basins of the world ocean[45]. This challenge has been pointed out in a recent study using global Ocean General Circulation Model (OGCM) simulations with embedded tides[46] and it is the first time that it is demonstrated with the use of *in-situ* velocity measurements. Notice that the $L_t$ values derived from the present study are based on the ADCP-measured velocity averaged in the 40–100 m layer. As both the geostrophic and internal wave motions are depth-dependent, the exact $L_t$ values relevant for delineating the SWOT-measured SSH signals remain to be determined.

In addition to its relevance to the SWOT mission, our above results have implications for submesoscale mixing. It is known that a part of the KE in the submesoscale range (mostly in the upper scale range) cascades towards larger scales and the other part (in the lower scale range) cascades towards smaller scales and is dissipated ultimately[7,8,47]. The coupling between internal waves and submesoscales may significantly affect the scale extension of these inverse and direct cascades and, therefore, the route to dissipation[47]. Also, as the submesoscale internal waves are ineffective stirrers for passive and active oceanic tracers[48], the regionally varying transition between the geostrophic flows and internal waves can likely affect the distribution of upper ocean properties. For an improved

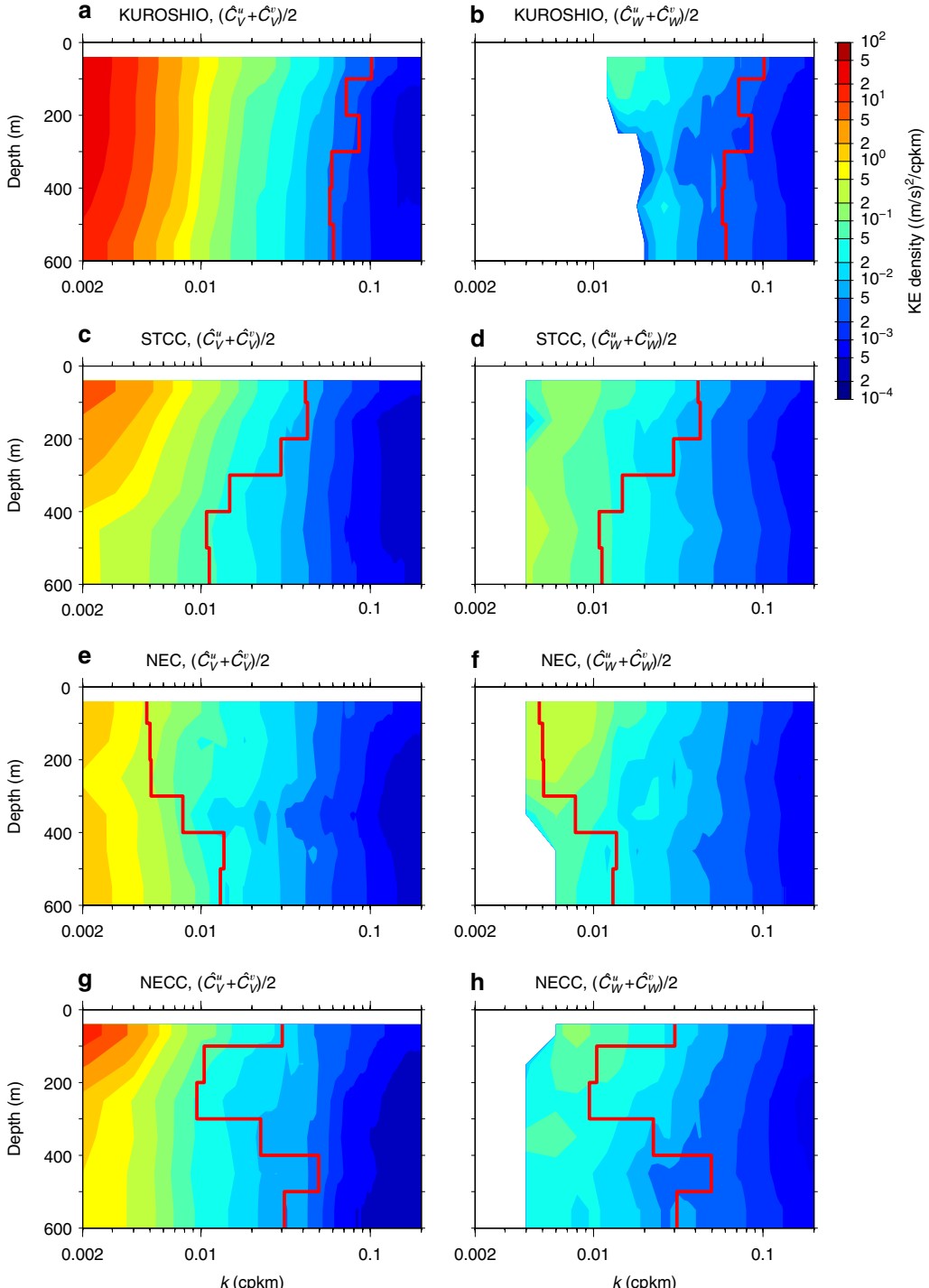

**Figure 6 | Transition scale dependence on depth.** Kinetic energy spectra for (**a,c,e,g**) balanced geostrophic versus (**b,d,f,h**) unbalanced wave motions in the (**a,b**) Kuroshio, (**c,d**) STCC, (**e,f**) NEC and (**g,h**) NECC band, as a function of depth. Red lines denote the transition scale $L_t$ where the geostrophic motion spectral amplitude matches that of the internal waves.

determination for $L_t$, it is important to emphasize the need for upper ocean buoyancy observations to accompany the velocity measurements. Rather than relying on the Garrett–Munk spectrum, concurrent buoyancy measurements will allow us to directly quantify the unbalanced wave motions in the wave–vortex decomposition[15,33]. Although the exact $L_t$ values will improve with future observations, we do not expect its geographical dependence as summarized by the red curves in Figs 5 and 6 to change qualitatively. By applying the analysis

approaches used in this study to shipboard ADCP (and upper ocean buoyancy) surveys in other parts of the world ocean, it will be important for future studies to establish the quantitative relationship between the $L_t$ values and the characteristics of regional mesoscale turbulence.

## Methods

**JMA ADCP data.** JMA utilizes two research vessels, *Ryofu Maru* and *Keifu Maru*, for its repeat hydrographic and shipboard ADCP surveys along 137°E (refs 19,20).

Hull-mounted 75 kHz broadband ADCPs from Teledyne RD Instruments were used before spring 2010, and they were switched to 38 kHz Ocean Surveyor ADCPs thereafter. A total of 33 transects from the period of 2004–2016 are selected and analysed in this study. Among these transects, 25 cover the full 3–34°N range as indicated in Fig. 1 and eight shorter transects span from 24 to 34°N (Fig. 1b). Seasonally, 10, 5, 10 and 8 transects are occupied in winter (January–February), spring (April–May), summer (June–August) and fall (October–December), respectively.

The raw ship-board ADCP data are processed using the Common Ocean Data Access System available from http://currents.soest.hawaii.edu/docs. As in previous studies utilizing the ship-board ADCP data[13–16], we assume that the ship velocity, averaged at 5 m s$^{-1}$, is much faster than evolving mesoscale and submesoscale features of our interest. The processed ADCP data are averaged in 5-min (∼2.2 km) horizontal and 16-m (or 20-m) vertical bins. With the SWOT mission in mind, we will primarily in this study focus on the near-surface $u$ and $v$ velocity averaged in the 40–100 m depth.

**Velocity spectral analyses.** Within each band of analysis, available cross-track (zonal) and along-track (meridional) velocity data are grouped into 500 km-long segments with 50% overlap, a Hanning window is applied to each segment, and a discrete Fourier transform is computed. The finalized along-track wavenumber spectrum for velocity variance is evaluated based on the squares of Fourier components averaged over all available segments.

In addition to the wavenumber spectra for the zonal and meridional current kinetic energy (KE), $\hat{C}^u/2$ and $\hat{C}^v/2$, we follow refs 15,16 and use the Helmholtz decomposition to calculate the KE spectra for the rotational and divergent flow components of the velocity field: $\hat{K}^\psi$ and $\hat{K}^\phi$. To do so, it is useful to introduce the following intermediary functions:

$$\hat{F}^\psi(s) = \frac{1}{2}\int_s^\infty \left[\hat{C}^u(\sigma)\cosh(s-\sigma) + \hat{C}^v(\sigma)\sinh(s-\sigma)\right]\mathrm{d}\sigma, \qquad (1)$$

and

$$\hat{F}^\phi(s) = \frac{1}{2}\int_s^\infty \left[\hat{C}^u(\sigma)\sinh(s-\sigma) + \hat{C}^v(\sigma)\cosh(s-\sigma)\right]\mathrm{d}\sigma, \qquad (2)$$

where $s = \ln(k)$ and $k$ is the along-track wavenumber. Given $\hat{C}^u/2$ and $\hat{C}^v/2$ estimated from the ADCP data, $\hat{F}^\psi$ and $\hat{F}^\phi$ in equations (1) and (2) can be evaluated numerically with the conditions $\hat{C}^u/2 = \hat{C}^v/2 \to 0$ as $k \to \infty$. Once $\hat{F}^\psi$ and $\hat{F}^\phi$ are determined, the KE spectra for the rotational and divergent components are given by

$$\hat{K}^\psi(k) = \hat{F}^\psi - \hat{F}^\phi + \frac{\hat{C}^u}{2} \qquad (3)$$

and

$$\hat{K}^\phi(k) = \hat{F}^\phi - \hat{F}^\psi + \frac{\hat{C}^v}{2}. \qquad (4)$$

Deriving $\hat{K}^\psi$ and $\hat{K}^\phi$ also helps to further decompose the ADCP velocity field into its wave and vortex components:

$$\hat{C}^u = \hat{C}^u_V + \hat{C}^u_W \text{ and } \hat{C}^v = \hat{C}^v_V + \hat{C}^v_W, \qquad (5)$$

where subscript $V$ stands for the vortex, or balanced geostrophic, component and $W$ for the ageostrophic wave component. To find $\hat{C}^u_W$ and $\hat{C}^v_W$, we need to estimate $\hat{F}^\psi_W$ and $\hat{F}^\phi_W$ first. To do so, we assume the divergent component of velocity is solely induced by the wave field and in this case $\hat{F}^\phi_W = \hat{F}^\phi$. By further assuming that the wave field can be parameterized by the Garrett-Munk spectrum[30], the ratio $\frac{f_0^2}{\omega_*^2} = \frac{\hat{F}^\psi_W}{\hat{F}^\phi_W}$ can be approximated by

$$\frac{f_0^2}{\omega_*^2} = \frac{\iint \frac{f_0^2}{\omega^2} l^2 \hat{C}^\phi(k,l,\omega)\mathrm{d}l\mathrm{d}\omega}{\iint l^2 \hat{C}^\phi(k,l,\omega)\mathrm{d}l\mathrm{d}\omega}, \qquad (6)$$

where $l$ is the across-track wavenumber and $\hat{C}^\phi(k,l,\omega)$ the Garret-Munk spectrum. Making use of $\hat{F}^\psi_W = \frac{f_0^2}{\omega_*^2}\hat{F}^\phi$, $\hat{C}^u_W$ and $\hat{C}^v_W$ can be estimated as follows:

$$\frac{\hat{C}^u_W}{2} = \left[1 - k\frac{\mathrm{d}}{\mathrm{d}k}\left(\frac{f_0^2}{\omega_*^2}\right)\right]\hat{F}^\phi - \frac{f_0^2}{\omega_*^2}\left(\hat{F}^\psi - \frac{\hat{C}^v}{2}\right) \qquad (7)$$

and

$$\frac{\hat{C}^v_W}{2} = \frac{f_0^2}{\omega_*^2}\hat{F}^\phi - \hat{F}^\psi + \frac{\hat{C}^v}{2}. \qquad (8)$$

With the wave component spectra $\hat{C}^u_W$ and $\hat{C}^v_W$ estimated, the geostrophic component spectra are simply $\hat{C}^u_V = \hat{C}^u - \hat{C}^u_W$ and $\hat{C}^v_V = \hat{C}^v - \hat{C}^v_W$.

One important assumption used in the Helmholtz and wave-vortex decompositions is horizontal isotropy. To assess this assumption for the time-varying, ADCP-derived, zonal and meridional velocities of $u'$ and $v'$, we follow ref. 49 and calculate the eddy anisotropic ratio $L/K$, where $K = (\overline{u'^2} + \overline{v'^2})/2$, $L = \sqrt{M^2 + N^2}$, $M = (\overline{u'^2} - \overline{v'^2})/2$, $N = \overline{u'v'}$, and over bars indicate time averaging. Averaged in the Kuroshio, STCC, NEC and NECC regions, the $L/K$ ratio is 0.38, 0.26, 0.24 and 0.52, respectively. These relatively high ratios are, however, dominated by mesoscale variability with wavelengths longer than 100 km. Within

the 5–100 km range of our interest (where mesoscale transitions from geostrophic motions to internal waves take place), these ratios reduce to 0.14, 0.08, 0.10 and 0.12, respectively. Given that these ratios are all under 15%, we believe the isotropic assumption underlying the Helmholtz and wave–vortex decompositions is justified in our study.

**Data availability.** JMA maintains a public website http://www.data.jma.go.jp/gmd/kaiyou/db/vessel_obs/data-report/html/ship/ship_e.php that includes the repeat ship-board ADCP data along 137°E. The intermediate data files and computing codes used in this study are available on request to the authors. Matlab is used in generating all figures.

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

## Acknowledgements

We are grateful to the past and present captains and crews of *R/V Ryofu Maru* and *Keifu Maru*, and staff of the Marine Division, Japan Meteorological Agency, for their laudable long-term observational efforts. We thank Jules Hummon and Eric Firing for help with the processing and deciphering of the shipboard ADCP data, and Lee Fu and Jinbo Wang for their in-depth discussions. B.Q. and S.C. acknowledge support from NASA SWOT and OSTST missions (NNX16AH66G and NNX13AE51E). P.K. acknowledges support of CNRS (France), LabexMer (ANR-10-LABX-19-01), as well as of the NASA-CNES SWOT mission.

## Author contributions

B.Q. and S.C. planned the research and wrote the initial manuscript. T.N. contributed to the data collection and analyses. P.K. provided guidance for the data analyses and interpretations. All authors contributed to the writing of the finalized manuscript.
