## [Peer Review File · Nature Communications]

Reviewers' comments:

Reviewer #1 (Remarks to the Author):

I recommend this paper for publication. It is a timely study of high-quality field data using recently developed analysis techniques. The perspective is on the upcoming SWOT campaign and the paper does a nice job indicating some of the challenges that will have to be overcome in order to disentangle balanced and unbalanced flows in the context.

Still, in the context of ship-based high-resolution measurements I recommend adding some discussion about the pressing need for buoyancy observations to accompany the velocity observations. The new wave-vortex decomposition method of BCF14 then allows comparing the predicted wave energy level to the total observed energy level, which is a very valuable analysis tool. Using the GM spectrum as a proxy for the buoyancy information, as done in the paper, is a comparably blunt tool. Not least because it is not clear (at least not to me) whether the GM spectrum is applicable in the first 100 meters of the ocean, as is relevant here. As I said, it would be useful to discuss this point a little.

Reviewer #2 (Remarks to the Author):

See attachment

Reviewer #3 (Remarks to the Author):

This paper reports a wavenumber-spectral assessment of measured surface velocities along a meridional section in the tropical and subtropical Northwestern Pacific. Its goal is a balanced currents vs. internal waves decomposition using the method in Refs. 14-15. The latter makes rather strong assumptions --- that the balanced currents are geostrophic, that the waves are linear, and that the flows are horizontally isotropic --- to be able to make a Helmholtz decomposition of the velocity vector from 1D data. So, in my view, the inferences from this method are far from certain. Nevertheless, the conclusions --- that balanced currents are more dominant at larger scales and in winter, that waves are more dominant at smaller scales and nearer the equator, and that the transition length scale varies geographically mainly in accord with the balanced kinetic energy level --- are probably qualitatively valid. These perspectives have largely been anticipated in previous studies elsewhere (Refs. 29 and 32), so the principal contribution here is for a different data set and a partly different geographical location (n.b., Ref. 29 is a global analysis of spectrum slopes, with shallower slopes interpretable as due to more waves).

We are pleased to know that the reviewers have found our investigations timely and important both for the SWOT satellite mission and for the physical oceanography discipline in general. We thank the reviewers for their constructive and concrete suggestions that have helped us improve our discussion and provide a better context for our analyses. We have revised the manuscript and the following summarizes the revisions and our responses.

Reviewer 1's comments:

I recommend this paper for publication. It is a timely study of high-quality field data using recently developed analysis techniques. The perspective is on the upcoming SWOT campaign and the paper does a nice job indicating some of the challenges that will have to be overcome in order to disentangle balanced and unbalanced flows in the context.

Still, in the context of ship-based high-resolution measurements I recommend adding some discussion about the pressing need for buoyancy observations to accompany the velocity observations. The new wave-vortex decomposition method of BCF14 then allows comparing the predicted wave energy level to the total observed energy level, which is a very valuable analysis tool. Using the GM spectrum as a proxy for the buoyancy information, as done in the paper, is a comparably blunt tool. Not least because it is not clear (at least not to me) whether the GM spectrum is applicable in the first 100 meters of the ocean, as is relevant here. As I said, it would be useful to discuss this point a little.

We agree with the reviewer that concurrent upper ocean buoyancy measurements, in addition to the ADCP-derived velocity measurements, are desired in future observations in order to better quantify the KE spectra of the unbalanced wave motions. This point is now emphasized in line 263–266 of the revised manuscript.

Reviewer 2's comments:

This manuscript investigates the upper-ocean dynamics across mesoscales to submesoscales (10-500 km). Specifically, the authors study the transition from geostrophic motions to internal waves using velocity measurements from a repeat ship transect in the northwestern Pacific (137E; 3-35N). By applying a Helmholtz and wave-vortex decomposition to one-dimensional wavenumber spectra of kinetic energy, the authors show that the transition has a strong geographic variability, likely associated with the mesoscale kinetic energy level.

This is a timely observational study on an important topic in physical oceanography. The novelty is the application of the Helmholtz and wave-vortex decomposition to a single dataset spanning different dynamical regimes. The results vindicate and extend previous

observational analyses that (i) together suggest significant geographic variability of the transition from geostrophic flows to internal waves (Buhler et al., 2014; Rocha et al., 2016) and (ii) present evidence of vigorous seasonality of submesoscale turbulence (Callies et al., 2015).

I believe the authors have material for a Nature Communications paper, but I urge them to rethink the context of their study, to try to generalize their findings, and to improve their prose. As the authors emphasize, an immediate application of their findings is the planning of the SWOT satellite mission and interpretation of SWOT data. But exclusive focus on SWOT makes the manuscript narrow in scope — with little effort the authors could make these results attractive to oceanographers and climate scientists beyond SWOT’s community.

1. The authors must discuss the assumptions of the Helmholtz and wave-vortex decompositions, which is the foundation of their analysis. A main concern is the assumption of horizontal isotropy, particularly near the Kuroshio and the NECC. How does failure of that assumption affect your results? [Potentially useful reference: Stewart et al. (2015).]

We agree with the reviewer that horizontal isotropy was assumed in our Helmholtz and wave-vortex decompositions of the ADCP velocity data. To assess this assumption, we followed Stewart et al. (2015) and found that the eddy anisotropic ratio L/K in the Kuroshio, STCC, NEC and NECC regions are 0.38, 0.26, 0.24, and 0.52, respectively, based on the ADCP velocity data. These relatively high ratios are, however, dominated by mesoscale variability with wavelengths longer than 100 km. Within the 5–100km wavelength range of our interest (where submesoscale transition from geostrophic flows to internal waves takes place), the L/K ratios reduce to 0.14, 0.08, 0.10, and 0.12, respectively. Given these ratios are all under 15%, we believe our adopted isotropic assumption is justified. In the revised manuscript, we have now included these comments in line 333–343.

2. The authors should discuss the vertical dependence of the spectra and their partition into geostrophic flows and internal waves. Figure 2(b) shows that the eddy motions are vertically inhomogeneous in the upper 300 m outside the Kuroshio. Do the spectra show a systematic depth dependence? Does the transition scale change with depth? This analysis can help you further characterize the geostrophic dynamics (see remark #4) — for example, SQG flows have a strong scale-dependent depth dependence that yields a systematic steeping of the kinetic energy (KE) spectrum with depth (Callies and Ferrari, 2013).

We thank the reviewer for raising this issue. In the revised manuscript, we have added a new Figure 6 in which KE spectra for the geostrophic and internal wave motions are plotted as a function of depth in the four flow regimes. Relevant discussion related to the SQG dynamics and reference to Callies and Ferrari (2013) are included; see line 219–238.

3. The authors should more clearly frame their study as a characterization of the transition from geostrophic motions to internal waves in the upper ocean. In hindsight, it is expected that geostrophic eddies and internal waves project onto similar length scales. There is no clear horizontal scale separation between geostrophic eddies and the internal wave con-

tinuum—there is only temporal scale separation between those flows. So there might be a transition between these two regimes. The fundamental questions are (i) what is the transition scale? and how the transition scale relates to the properties of the geostrophic flow?

(a) How does the transition scale L_c vary with the kinetic energy (KE). I realize you do mention this relationship, but you could plot L_c vs. $\int E(k)dk$, where $E(k)$ is the KE spectrum.

(b) How does the transition scale L_c relate to the energy-containing scale? You could estimate the energy-containing scale from the KE spectrum and plot L_c vs. L_e — you could ask Callies and Rocha for the L_c and L_e values in the Gulf Stream, Eastern Pacific, and Drake Passage, so that you have more data points.

(c) Is there a relationship between L_c and $\int E(k)dk$, and L_c and L_e ? Can you use those relationships to produce a global map of the transition scale L_c based on L_e and $\int E(k)dk$ estimated from current altimeters? You will have to assume that the kinetic energy of the waves is uniform, as you seem to suggest in your discussion starting in line 191. Such a global estimate of L_c will be very rough, and it is likely to comfort and discomfort equally. In any event, the simple analysis (a)-(b) may help you synthesize your findings.

(d) Callies et al. (2014) suggest that there is a similar transition in the atmosphere. This uncanny similarity is poorly understood — it is worth a comment in your discussion.

We appreciate very much reviewer’s concrete suggestions. To better emphasize the “transition” scale from geostrophic motions to internal waves, we have changed L_c (the “crossing” scale) to L_t throughout the revised text.

For (a), we have plotted in the revised manuscript L_t vs. $\int E(k)dk$ as Figure 5c. The linear correlation between L_t and $\int E(k)dk$ in Figure 5c is not very high at $r = -0.44$. Interestingly, and in connection to reviewer’s point (b), L_t has a higher linear correlation, $r = -0.56$, with the energy-containing scale L_e . This is emphasized by new Figure 5d and the discussion in line 201–218 in the revised manuscript.

Given the length of our manuscript submitted as a *Nature Communications* article, we decided not to expand the L_t estimation globally in the present study. We appreciate reviewer’s comment (c) and plan to pursue the global estimation for L_t in a follow-up study to this article.

With regard to item (d), we thank the reviewer for bringing our attention to Callies et al. (2014). We have now referenced this study and noted that a similar transition occurs in the mid-latitude atmosphere as well (line 126–128).

4. What is the nature of the geostrophic variability in the different regimes? You seem to be interpreting the geostrophic flows as a combination of “interior” quasigeostrophic (QG) turbulence and QG turbulence driven by mixed-layer instabilities. While this is likely true in the Kuroshio and STCC, I’m unconvinced that’s the case in the NEC and NECC. Can SQG turbulence or wavy linear QG dynamics be important in those regions? [Potentially useful references: Tulloch et al. (2009, 2011).]

Our previous studies indicated that the meridional PV gradient in the NEC of the northwestern Pacific lacks sign reversal and, as a consequence, it is baroclinically stable (Ref. 24). This result is consistent with the findings by Tulloch et al. (2009). The NECC, on the other hand, is subject to intense barotropic instability due its strong lateral shear (Ref. 28). In the revised manuscript, we have now referenced Tulloch et al.'s work and have made the above points explicit; see line 81-89 and 204-207.

5. The authors could discuss the implication of their results to submesoscale lateral mixing in the upper ocean. I believe the implications of your results for submesoscale mixing in the upper ocean is more fundamental than the application for the planning of SWOT — internal waves are ineffective stirrers. So the regionally varying transition between geostrophic flows and internal waves might affect the distribution upper ocean properties, etc. (cf lines 42-46).

Yes, we agree with the reviewer and have now included the implication of our results for submesoscale lateral mixing in the upper ocean; see line 258-263.

Minor points

1. Lines 34-36: To my knowledge there is no theory of “interior QG turbulence with mixed layer instabilities” — as the authors mention, there is numerical evidence of those flows co-existing, but that’s far from a theoretical prediction — add a reference or clarify.

The original sentence in line 134-136 was not well constructed. What we meant to say was that the observed spectral slope of $k^{-2.5}$ fell in between k^{-3} and k^{-2} , the slopes predicted for the interior QG turbulence and the mixed layer instability, respectively. In the revised manuscript (line 132-134), we have modified the sentence and added the references.

2. Lines 38-39: add a reference to support the claim that mesoscale eddies account for 90% of the eddy KE of the ocean.

Reference to Ferrari and Wunsch (2009) is now added.

3. Lines 43-44: meaning of “mixed layer evolution” is ambiguous — clarify.

We have changed “mixed layer evolution” to “mixed layer development”.

4. Lines 47-48: only on a regional scale.

Yes, we agree and have made this point explicit.

5. Lines 201-205: this seems to be consistent with the original “saturation hypothesis” of Garrett-Munk. Also, it’s not obvious that the results for the deep ocean (Ref. 34) are valid for the upper ocean — clarify.

We have mentioned in the revised manuscript that the result of Figure 5b is consistent with Garrett-Munk’s saturation hypothesis. The “deep ocean” used in the title of Sugiyama and Hibiya (Ref. 37) actually indicates the open ocean away from lands. Their results include all vertical modes of internal waves and are hence relevant for the upper ocean.

6. Line 217: after looking at figures 3 or 4 or 5b, I realize that the spectrum of the waves is not recovered at large scales owing to inaccuracies of the Helmholtz decomposition, but I believe your remark about the “commonly valid range” will puzzle most readers — clarify.

We agree that the wording “commonly valid range” is confusing. It is now removed from the text.

7. The NECC band: Figures 1a and 2b suggest that the NECC region eddy KE is as large as the Kuroshio region eddy KE, yet the transition scale is much larger in the NECC. This suggests that the transition scale is determined not only by the KE level, but also by how fast the KE spectrum decays, among other things — expand your discussion.

The linear color scale used in our original Figures 1a and 2b was not adequate to differentiate the maximum EKE level in the Kuroshio and NECC regions. In the revised manuscript, we have changed the linear scale to a log scale and the EKE level in the Kuroshio can be seen clearly now to be larger than that in the NECC. Please see revised Figure 2b, as well as the newly-added Figure 5c.

8. Seasonality in the Kuroshio: Why is the seasonality of the KE spectrum of the geostrophic flows restricted to scales smaller than about 80 km?

Our recent high-resolution simulation studies indicated that the seasonal spectral difference is largely due to the occurrence of mixed layer instability in winter, which tends to produce eddy signals with scales shorter than 80km. In the revised manuscript, we have made this point now explicit; see line 164–168.

9. Terminology: consider using “transition scale” in place of “crossing scale” or “scale that separates the dominance (...)”; also consider using “geostrophic” instead of “balanced” and “inertia-gravity waves” as opposed to “unbalanced wave motions.”

Yes, we adopted the “transition scale” in the revised manuscript. “Geostrophic” and “inertia-gravity waves” are also used now throughout the manuscript.

10. Title: consider using a spunky title, e.g., “Submesoscale transition from Geostrophic Flows to Internal Waves in the Upper Ocean: Observations from the NorthWestern Pacific.”

We have modified the title as recommended.

11. Units: use SI units throughout. I understand there are some instances where the old

fashioned cm for length is convenient owing to small numerical values. But this is not the case in your paper.

Yes, we have changed all units to SI units.

12. Figures: consider improving the quality of your figures [e.g., look how well crafted are the figures in Callies et al. (2015).]

For quality of figures, we have changed the linear scale to log scale for Figures 1a and 2b and have adjusted the color shade. For Figures 3 and 4, we have unified the confidence levels by shade and modified all legends, so that they are now consistent with those used in the text. Finally, we have improved the colorbar in Figure 5.

13. Reproducibility: to my knowledge, Nature lacks a policy on data and code sharing. Nonetheless, I encourage you to share codes and intermediate data files. This may increase the reproducibility of your results.

All the ADCP data have been collected by the Japan Meteorological Agency and it maintains a public website: http://www.data.jma.go.jp/gmd/kaiyou/db/vessel_obs/data-report/html/ship/ship_e.php. In the “JMA ADCP data” section under Methods, we have now included this website and indicated that all intermediate data files and computing codes are available upon request to the leading authors, BQ and TN.

Remarks on the text

We want to thank the reviewer for numerous and detailed suggestions to improve the text and presentation. All those suggestions have been incorporated into the revised manuscript.

Reviewer 3’s comments:

This paper reports a wavenumber-spectral assessment of measured surface velocities along a meridional section in the tropical and subtropical Northwestern Pacific. Its goal is a balanced currents vs. internal waves decomposition using the method in Refs. 14-15. The latter makes rather strong assumptions — that the balanced currents are geostrophic, that the waves are linear, and that the flows are horizontally isotropic — to be able to make a Helmholtz decomposition of the velocity vector from 1D data. So, in my view, the inferences from this method are far from certain. Nevertheless, the conclusions — that balanced currents are more dominant at larger scales and in winter, that waves are more dominant at smaller scales and nearer the equator, and that the transition length scale varies geographically mainly in accord with the balanced kinetic energy level — are probably qualitatively valid. These perspectives have largely been anticipated in previous studies elsewhere (Refs. 29 and 32), so the principal contribution here is for a different data set and a partly different geographical location (n.b., Ref. 29 is a global analysis of spectrum slopes, with shallower slopes interpretable as due to more waves).

The reviewer's point about our assumption of horizontal isotropy is well taken. We have now addressed the validity of this assumption in the revised manuscript; please see our reply to Reviewer 2's comment #1. For the internal waves, we did not assume their linearity. As mentioned in the Methods section, we assume their spectrum follows that of Garrett-Munk, which is a consequence of strong nonlinear interaction. Our interest in the geostrophic motions stems from the relevance of our study to the future SWOT mission.

We agree with the reviewer that Xu and Fu (2009; Ref. 29) is a global analysis and their observed shallower SSH spectral slopes could, in hindsight, be interpreted as due to more internal wave motions. Because Xu and Fu (2009) used the SSH data alone, it was not possible for them to distinguish the contributions from the geostrophic versus internal wave motions. Indeed, the merit of the ADCP measurements is its information of concurrent u and v velocities that allows us to quantify these two motions via the Helmholtz and wave-vortex decomposition.

Our analysis was very much inspired by Callies et al. (2015; Ref. 32). Callies et al. (2015) have focused largely on the Gulf Stream, a region of high mesoscale eddy variability. Our study has applied the Helmholtz and wave-vortex decomposition to a repeat ADCP dataset that spans across different dynamical regimes. It is this spanning across different dynamical regimes that afforded us, for the first time, to examine the submesoscale transition from the geostrophic flows to internal waves as a function of geographical locations, seasons, and water depths. We hope our study will lead to future observational and modeling studies that will further improve our understanding of the geostrophic versus unbalanced wave motions.

The above are our replies to the reviewers' comments. Again, we appreciate very much the reviewers' constructive comments and thoughtful suggestions that have helped us improve our discussion and expand the manuscript's scopes.

REVIEWERS' COMMENTS:

Reviewer #2 (Remarks to the Author):

The authors have successfully addressed my technical concerns and minor points. Below I provide a list of typos and a few optional minor points. But I don't need to see this manuscript again.

I commend the authors on a fine job revising their manuscript and look forward to seeing this paper in *Nature Communications*.

1. Lines 56 and 111: The Bühler et al. (2014) method is a *spectral* Helmholtz decomposition — it only separates the statistics (KE spectra) not the velocity field. Consider adding the qualifier *spectral* to Helmholtz decomposition.
2. Line 98-101: This division in subdomains is necessary to comply with the horizontal (statistical) homogeneity. Meaningful wavenumber spectra (and their decompositions) exist only in the realm of homogeneous statistics. The theoretical spectral predictions the authors mention and the GM spectrum also assume lateral homogeneity.
3. Lines 161-164: Consider re-writing (simplifying) the confusing sentence “Rather than (...)”
4. Line 200: Consider adding the qualifier *internal* to tides to avoid ambiguity.
5. Lines 258-264: I appreciate the discussion about the implications for submesoscale mixing. But I'm unconvinced the authors need to ground their discussion on forward and inverse cascades, etc. Most of the submesoscale lateral stirring is likely performed by geostrophic flows (internal waves are ineffective stirrers). Because the nature of the stirring (local vs. non-local; Scott, 2006) depends on the spectral slope, one may need to remove the internal wave component first, before making inferences about mixing — e.g., trying to understand the nature of submesoscale stirring based on the (total) spectra of the NECC may lead to misleading interpretations since internal waves account for the high-wavenumber flattening of the spectra.
6. Figure 1a: Consider demarcating the four regions (similarly to figure 2b).
7. Typos
 - (a) Line 50: “(...) ~~research vessels sail~~ (...)” → “(...) research vessels *that* sail (...)”
 - (b) Line 88: ~~measoseale~~ → mesoscale.
 - (c) Line 187: “(...) ~~in~~ the right column.” → “(...) *on* the right column.”
 - (d) Line 224: “(...) ~~weak dependence~~ (...)” → “(...) weak *depth* dependence (...)”

References

Scott, R. (2006). Local and nonlocal advection of a passive scalar. *Physics of Fluids* 18(11), 116601.

Reply to Reviewer 2 of the manuscript NCOMMS-16-19172A

We are pleased to learn that Reviewer 2 has found that we have successfully addressed his/her concerns and points. In finalizing the manuscript, we have incorporated Reviewer 2's new comments as follows:

1. Lines 56 and 111: The Buhler et al. (2014) method is a spectral Helmholtz decomposition – it only separates the statistics (KE spectra) not the velocity field. Consider adding the qualifier spectral to Helmholtz decomposition.

We agree with the reviewer and have added the qualifier “spectral”.

2. Line 98-101: This division in subdomains is necessary to comply with the horizontal (statistical) homogeneity. Meaningful wavenumber spectra (and their decompositions) exist only in the realm of homogeneous statistics. The theoretical spectral predictions the authors mention and the GM spectrum also assume lateral homogeneity.

We agree and have commented on the need to divide in subdomains to ensure horizontal homogeneity in line 101-102 of the revised manuscript.

3. Lines 161-164: Consider re-writing (simplifying) the confusing sentence “Rather than (...).”

We have modified the sentence in line 163-166.

4. Line 200: Consider adding the qualifier internal to tides to avoid ambiguity.

We have added the suggested qualifier.

5. Lines 258-264: I appreciate the discussion about the implications for submesoscale mixing. But I'm unconvinced the authors need to ground their discussion on forward and inverse cascades, etc. Most of the submesoscale lateral stirring is likely performed by geostrophic flows (internal waves are ineffective stirrers). Because the nature of the stirring (local vs. non-local; Scott, 2006) depends on the spectral slope, one may need to remove the internal wave component first, before making inferences about mixing – e.g., trying to understand the nature of submesoscale stirring based on the (total) spectra of the NECC may lead to misleading interpretations since internal waves account for the high-wavenumber attenuing of the spectra.

We agree with the reviewer's point and have mentioned explicitly now (line 266-268) that the submesoscale internal waves are ineffective stirrers for passive and active oceanic tracers. Reference to Scott (2006) is now newly included.

6. Figure 1a: Consider demarcating the four regions (similarly to Figure 2b).

We have added the demarcating lines as recommended.

7. Typos

We have corrected the typos identified by the reviewer. Thank you very much.